# Microsystem Advances through Integration with Artificial Intelligence

**DOI:** 10.3390/mi14040826

**Published:** 2023-04-08

**Authors:** Hsieh-Fu Tsai, Soumyajit Podder, Pin-Yuan Chen

**Affiliations:** 1Department of Biomedical Engineering, Chang Gung University, Taoyuan City 333, Taiwan; d1131005@cgu.edu.tw; 2Department of Neurosurgery, Chang Gung Memorial Hospital, Keelung, Keelung City 204, Taiwan; 3Center for Biomedical Engineering, Chang Gung University, Taoyuan City 333, Taiwan

**Keywords:** microfluidic device, lab-on-a-chip, micro total analysis system, artificial intelligence, machine learning, personalized medicine

## Abstract

Microfluidics is a rapidly growing discipline that involves studying and manipulating fluids at reduced length scale and volume, typically on the scale of micro- or nanoliters. Under the reduced length scale and larger surface-to-volume ratio, advantages of low reagent consumption, faster reaction kinetics, and more compact systems are evident in microfluidics. However, miniaturization of microfluidic chips and systems introduces challenges of stricter tolerances in designing and controlling them for interdisciplinary applications. Recent advances in artificial intelligence (AI) have brought innovation to microfluidics from design, simulation, automation, and optimization to bioanalysis and data analytics. In microfluidics, the Navier–Stokes equations, which are partial differential equations describing viscous fluid motion that in complete form are known to not have a general analytical solution, can be simplified and have fair performance through numerical approximation due to low inertia and laminar flow. Approximation using neural networks trained by rules of physical knowledge introduces a new possibility to predict the physicochemical nature. The combination of microfluidics and automation can produce large amounts of data, where features and patterns that are difficult to discern by a human can be extracted by machine learning. Therefore, integration with AI introduces the potential to revolutionize the microfluidic workflow by enabling the precision control and automation of data analysis. Deployment of smart microfluidics may be tremendously beneficial in various applications in the future, including high-throughput drug discovery, rapid point-of-care-testing (POCT), and personalized medicine. In this review, we summarize key microfluidic advances integrated with AI and discuss the outlook and possibilities of combining AI and microfluidics.

## 1. Introduction

The study and control of fluids in channels and systems with perimeters generally between a few micrometers to a few millimeters is known as microfluidics [1]. The microfluidic flows are typically of low Reynolds number, i.e., viscous forces outweigh inertial forces, and are thus laminar in nature [2]. The high surface-area-to-volume ratio in microfluidics enables faster heat and mass transport, which enhances reaction control and speeds up reaction time [3]. The microfluidics discipline entails the theoretical and technological development of accurate manipulation of microscopic fluid volume and examination of phenomena occurring within it. Applications using microfluidic devices can be found in broad domains across biomedicine, chemistry, and material science [4,5,6].

A branch of artificial intelligence called machine learning is concerned with creating algorithms that can learn from data and establish the association between data and the features of interest [7]. Moreover, the iterative update of parameters in machine learning algorithms allows their accuracies to increase when the algorithms “learn” from additional data. Furthermore, development of artificial neural networks (ANNs) has advanced artificial intelligence in recent decades, particularly in image and language recognition [8,9]. In neural networks, influenced by applied mathematics, physics, and neuroscience, artificial neurons are mathematically modeled and arranged in layers to process information similar to how biological neurons would be. During training, the weight of each association between neurons is updated by backward propagation to establish the association between data and features [10]. In deep learning, as the neural network layers deepen and the number of neurons increases, more data are required, but patterns in complex data can be recognized [11]. As a result, the training of it becomes more computationally demanding, but through the advances of parallel processing offered in the graphics processing unit (GPU), the field of deep learning continues to thrive and has seen innovation to various applications such as self-driving car, security, drug development, and medical image diagnosis [12,13,14,15,16,17].

Intelligent microfluidics, a field enabled by the integrating the data processing capability of machine learning and data science with the automation potential in microfluidics [18], is particularly effective in advancing quantitative biology and rapid medical diagnostics [19]. Next-generation biomimetic microfluidic systems such as organ-on-chips (OoCs) have great potential to recapitulate and model human physiological as well as pathophysiological processes in vitro. The complex and vast amounts of high-throughput data generated on these platforms are also great opportunities for AI.

In this review, we discuss the development, applications, and outlook of intelligent microfluidics, where advancements are made through integration between microfluidics and artificial intelligence (Figure 1).

## 2. AI-Enhanced Smart Platform and Automation

### 2.1. Computer-Aided Microsystem Design and Optimization

Microfluidic chips take advantage of physical phenomena at smaller length scales; therefore, miniaturization of these devices often requires customized design. Recent advances in machine learning have been applied to the design and optimization of microfluidic systems [20].

The entire hydraulic circuit and components within it can be analyzed and optimized using circuit modeling [21,22,23,24,25]. Several machine learning and numerical simulation methods have been applied to microdevice design for flow sculpturing [26,27,28], micromixing [29,30,31,32], microjetting [33], droplet microfluidics [34,35], and extensional rheometry [36]. For example, a convolutional neural network (CNN) is trained to learn the sculpturing outcome of flow due to micropost locations in the microfluidic channel; the system has improved performance over conventional computational fluid dynamics (CFD)-based forward models and can potentially predict results when the condition is outside the scope of what was used in the training (Figure 2a) [27]. By taking the AI approach, design iteration of microfluidic chip can be rapidly accelerated.

Moreover, machine learning techniques can also be applied to the fabrication of microsystems. For example, Wang et al. adapted a fully automated CNN computer vision system to aid in calibration during 3D printing of microstructure dimensions [37]. Shchanikov et al. used an ANN to design a bidirectional biointerface with nanoelectronics and microfluidics [38]. Contemporary numerical simulation software can readily integrate with machine learning techniques to iterate microdevice designs, accelerating device development and validation, e.g., the ones used for particle trajectory prediction can be adapted for predicting red blood cell (RBC) movements in microchannels [39,40,41].

#### Computational Fluid Dynamics (CFD) Modeling

In recent years, advances in GPUs have made the training of deep ANNs possible. The features in structured data sequences, images, and videos can be quickly learned and inferred in modeling fluid dynamics [42,43,44].

In continuous-flow microfluidics, Cai et al. introduced artificial intelligence velocimetry (AIV), which integrated imaging data with physics-informed neural networks, to measure the velocity and stress fields of blood flow [45]. By utilizing neural networks to automatically evaluate experimental data and infer important hemodynamic markers that quantify vascular damage, AIV offered a novel paradigm that seamlessly blended pictures, experimental data, and underlying physics. Similarly, the flow velocity profile in a shallow microfluidic channel could also be extracted from the scalar signal transport through a deep neural network [46]. In microfluidic systems, often a system is in the laminar regime with low Reynolds number and high Péclet number, which means that viscous force and advection are more dominant than effects from inertia and diffusion. In certain applications, such as concentration gradient generation and chemical synthesis, mixing efficiency in microchannels becomes important. The fluid behavior in microstructure-based passive microfluidic mixers can be predicted using AI, and designs can be rapidly iterated in comparison to conventional physics-coupled numerical simulations where flow field and chemical transport has to be sequentially solved [29,30,31,32,47].

Multiphase flow is also a discipline that has many applications, such as two-phase emulsion, protein extraction in biotechnology, and injection molding in manufacturing. The flow pattern in two-phase fluid mixing can be rapidly modeled using neural networks trained with physical knowledge and comparable to that by classical CFD solutions, where approximation to Navier–Stokes equations requires large amount of computational resources [48,49,50,51]. Moreover, combinatorial multiphase flow can be utilized to assemble colloidal materials in microfluidics, and machine learning can accelerate the predictability of material characteristics for designing such materials [52]. An AI algorithm to predict multiphase fluid dynamics can accelerate and aid in the design of microfluidic chips for these applications.

### 2.2. Automation Control in Microsystems

#### 2.2.1. Flow Control

The flow in microfluidics can be actuated by pumps of different mechanisms, such as syringe-driven, peristaltic, pressure-driven, piezoelectric, electro-osmotic, or microvalve-based peristaltic micropumps [53]. Dynamic laminar flow control by pistons or syringe pumps is required in applications where joining flow boundary location requires precise control and syncing. Dressler et al. developed a reinforcement learning-based deep Q network that utilized image feedback for rapid and high accuracy flow control and droplet size control [54].

Micropumps composed of microvalves or piezoelectric valves often become important components due to their ability to integrate into the microdevice and to realize the development of the micro-total analysis system (μTAS) [55]. Abe et al. used reinforcement learning for timing control of valve operation in peristaltic micropumps to increase the maximum flow rate [56,57]. Integration of multiple microvalves into a fully programmable biochip was designed by Shayan et al. [58]. Integration of machine learning techniques with the fully programmable platform enables real-time detection of biochip status and detection of potential attacks against a real-world bioassay.

#### 2.2.2. Thermal Control

The advances in microelectromechanical systems (MEMS) continue to decrease the size and increase the power, therefore thermal management on MEMS devices becomes a vital component to the system. Due to the small length scale and small thermal resistances in microfluidics, a microfluidic heat sink was applied for cooling MEMS chips [59]. Moreover, precise heating or cooling of microfluidics is also important for applications in μTAS [60].

High speed and accurate thermal control of a microfluidic platform is made possible by the use of an ANN trained on infrared (IR) thermography [61]. Quinn et al. also reported an optically controlled thermofluidics platform with machine learning feedback control [62].

As an alternative to passive thermal measurement using IR thermography, temperature-sensitive species can be introduced to the system and report local temperatures in the microenvironment, such as nanocrystals or quantum dots. However, the temperature often needs to be inferred from photoluminescence data and the accuracy is poor. By applying a fully-connected ANN, Lewis et al. demonstrated that improved accuracy and temporal resolution of temperature in the local microenvironment were obtained using cadmium telluride (CdTe) quantum dots [63].

**Figure 2 micromachines-14-00826-f002:**
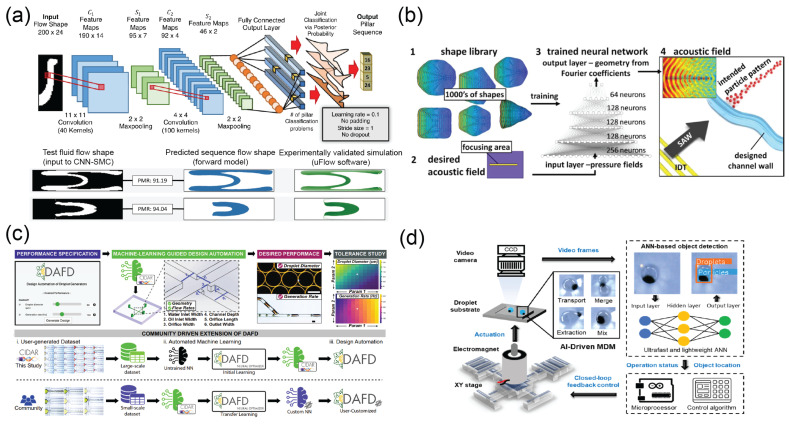
Selected examples of AI-enhanced smart platforms and automation. (**a**) A CNN framework to predict the outcome of flow sculpturing by micropillar microstructures [27]. Reproduced under Creative Commons Attribution 4.0 International license (CC–BY). (**b**) A DNN to predict particle patterning by surface acoustic waves in a shape library was proposed by Raymond et al. [64]. Reproduced under CC–BY license. (**c**) Machine-learning-empowered design automation of droplet microfluidics channel [34]. Reproduced under CC–BY license. (**d**) Machine-learning-assisted classification for droplet manipulation in a magnetic digital microfluidic platform [65]. Reproduced under CC–BY license.

#### 2.2.3. Particle Manipulation

In many applications, such as separation, diagnostics, and patterning, particle manipulation is often a required process. The low reagent requirements and smaller length scale make microfluidics a good candidate platform for such applications [66]. The particle manipulation can be exerted through electromagnetic, optical, hydrodynamic, or acoustofluidic techniques, while machine learning methods can be further adapted for precise and rapid control.

The particles in the fluid system can be manipulated through flow dynamics. Fang et al. used an ANN-based feedback pump control to create a hydrodynamic Stokes trap for particle manipulation [67]. Through reinforcement Q-learning, Abe et al. also demonstrated that a micropump-based system can learn to control and manipulate a microparticle to reside in a predefined target area [57]. Particles in a curved channel or in a straight channel filled with viscoelastic fluids also experienced inertial motion and were manipulated [68,69]. Recently, Su et al. and Hamdi et al. separately reported AI-accelerated approaches to calculate lift coefficient in microfluidic channels for inertial focusing based on knowledge from the computationally-expensive direct numerical simulation method [70,71].

Magnetic particles can also be conjugated to a desired biorecognition element, and an external magnetic field can be applied for magnetophoretic isolation [72]. Koh et al. combined magnetophoresis with supervised learning for sorting and purification of normal sperm based on the slender body theory [73]. Optical power of light can also exert forces on microscopic particles in the optical tweezers technique. While conventional control theory such as the proportional–integral–derivative controller is commonly adapted, machine learning techniques can also be used to increase speed and accuracy when calculating the optical force and manipulation [74,75].

In the field of acoustofluidics, where a piezoelectric actuator or platform is used, the particles can be spatially manipulated based on their acoustophysical properties, typically though a feedback control [76]. Raymond et al. trained a DNN on pictures of presolved acoustic fields to allow rapidly tailored acoustic fields for patterning microparticles in microchannel components (Figure 2b) [64]. By using AI to predict the acoustic field, the design of an acoustofluidics chip can be optimized and automated for tailoring specific applications. While conventional numerical and analytical solutions are used to resolve acoustic pressure fields in designed geometry, the inverse problem can be resolved more efficiently by taking the DNN approach. Similarly, Yiannacou et al. demonstrated that control of a single ultrasound transducer frequency through an epsilon-greedy algorithm can be used to control and manipulate the particles [77,78].

#### 2.2.4. Droplet Control, Detection, and Tracking

Droplet microfluidics is a particular case of multiphase microflow, where two or more immiscible fluids are split into single or multiple emulsions as tiny liquid reactors for bioanalysis, chemical synthesis, cell culture, and more [79,80].

To produce droplets of desired size, precise control and minimization of variation of flow rates are essential [54]. Mahdi and Daoud demonstrated that an ANN can be used to predict the size of microdroplets [81]. Lashkaripour et al. proposed a web-based software tool, DAFD, with machine learning algorithms to predict droplet formation performance with a high level of accuracy in a flow-focusing microfluidic chip (Figure 2c) [34]. The machine learning algorithm can also automate the microfluidic chip design for a specific droplet size desired by the user. In particular, the DAFD software was designed for scientific community collaboration, where the machine learning algorithms can learn both from the data in the developer’s laboratory and from volunteered data from the community. The federated learning approach can aid in creating a universal neural network for data collected in different settings [82]. Similarly, Siemenn et al. adapted a Bayesian inference with droplet size measurement feedback from images to optimize droplet generation [83]. The stability of microdroplets can be significantly affected by the type of surfactant and its concentration. Khor et al. established a convolutional autoencoder model to anticipate if a droplet would become unstable and break up [84]. Chagot et al. utilized a Bayesian regularized ANN and XGBoost algorithms to predict droplet size given flow rate and surfactant concentrations with mean absolute percentage error as low as 3.9%, in comparison to other state-of-the-art models [85].

Droplet segmentation and classification in images can be achieved using machine learning techniques as well. Various versions of CNN-based You Only Look Once (YOLO) object detectors have been used for automated detection of droplets and content within them at frame rates of up to 100 frames per second (FPS) [86,87]. Subsequent tracking of droplets to extract droplet dynamics is also practiced using deep learning techniques under brightfield and fluorescence microscopy [88,89].

In machine learning, the algorithm learns to establish the association between features and data. Therefore, a trained neural network can also predict the experimental conditional from image results [90]. For example, the microdroplet images can be predicted by a trained deep neural network for the flow rate or concentration used based on the droplet’s size and optical property, reaching accuracy within 0.5% [91]. Arjun et al. also reported that droplets with various chemical mixing efficiencies can be classified by a trained CNN [92]. Droplet freezing and extrapolation of exact temperature for ice nucleation on a microfluidic freezer platform was made possible using a DNN model with a polarized light detection [93]. Tang et al. reported an ANN to classify microdroplets containing magnetic particles and showed its manipulation with failure-correction capability, demonstrating the superiority of ANN compared to conventional magnetic digital microfluidic platform [65] (Figure 2d). The integration of AI-based image analysis in the automation and control process of the digital microfluidic platform provides additional advantages of quality assurance and online decision making.

After detection and classification of desired droplets, fluid routing or sorting can be integrated in the downstream microfluidic component to isolate them. Automated routing of droplets has been demonstrated using deep reinforcement learning [94] and evolutionary algorithms [95]. Alternatively, droplets containing desired products such as cells or particles can be sorted as well when the AI for the detection is combined with an active component for target purification [96,97,98].

## 3. Process Optimization and Discovery

### 3.1. Synthetic Reaction Optimization

One advantage of AI is its ability to sequentially generate a large amount of experimental parameters for automated iteration, and it also discerns patterns in the result. Therefore, many applications exist that automate microfluidics with AI for large-scale experimental iteration in medicine, material science, and energy development [18,99].

For chemical synthesis, machine learning algorithms can be used to automate the experiments and optimize based on feedback when combined with robotics [100,101,102]. McMullen and Jensen demonstrated an automated microreaction system incorporated with silicon-based microfluidic devices and machine learning for optimization of synthesis without the need for *priori* information, i.e., a “black-box” optimization, to the reaction parameters [103,104] (Figure 3a), which means the microreaction system can be placed in continuously self-optimization to find the reaction parameters for best product properties set by the designer. Rizkin et al. used a machine learning algorithm with IR monitoring to automate microchemical reactors in the polymerization of zirconocene [105]. The parameters of the chemical kinetics of a complex reaction can also be modeled with the aid of machine learning, leading to a broad technoeconomic outcome.

### 3.2. Nanoparticle Synthesis

Control of flow rate, concentration, and mixing are vital variables for nucleation and synthesis of nanoparticles on the microfluidic platform and one of the applications adapting AI [106]. Krishnadasan et al. adapted a noise-tolerant global search algorithm for a black-box optimization of injection rate and temperature for synthesizing CdSe nanoparticles on a Y-shaped microfluidic reactor from CdO and Se precursor solutions [107]. Similarly, ANN can be used to extract the condition–property relationship from the combinatorial synthesis data and provide conditions to synthesize nanoparticles [108,109,110,111,112,113], nanotubes [114], nano-formulation of pharmaceuticals [115], quantum dots [116,117], liposomes [118], or polymeric microparticles [119,120] (Figure 3b). Diamiati et al. trained an ANN to predict particle size of poly(Lactide-co-glycolide) (PLGA) microparticles, a biocompatible drug delivery polymer, based on the conditions of the flow focusing synthesis experiments on multiple different synthesis platforms. The learnability or the trainability of the AI enables the in silico size-dependent design of microparticle synthesis for tailored applications.

Notably, the trainability of neural networks not only provides an optimization method for synthesis in that early stage when the data size is small but can also continue to refine through learning the new data and further feedback the optimization [110] (Figure 3c). It is expected that more machine learning, in particular generative neural networks, will be adapted in nanoparticle synthesis workflow in future for designing application-tailored nanoparticles in a variety of disciplines, such as medicine, energy, and more.

**Figure 3 micromachines-14-00826-f003:**
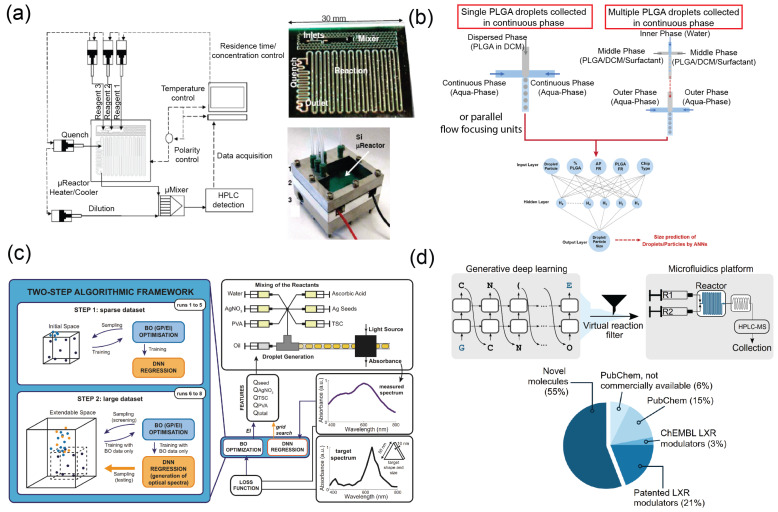
Microfluidics integrated with machine learning for process optimization and drug discovery. (**a**) An automated microfluidic system for automated optimization of chemical synthesis. Reprinted with permission from [103]. Copyright 2010 American Chemical Society. The font size have been adjusted for sake of clarity. (**b**) A droplet microfluidic platform combined with AI for synthesizing poly(Lactide-co-glycolide) microparticles from precipitating out of dichloromethane (DCM) [119]. Reproduced under CC–BY license. The figure has been reproduced for sake of clarity. (**c**) Two-step drug synthesis microfluidic platform with machine learning optimization [110]. Reproduced under CC–BY license. (**d**) A micromixer microfluidic platform combined with generative AI for de novo drug design and synthesis [121]. Reproduced under CC BY–NC (noncommercial) license.

### 3.3. Drug Development

The automation and miniaturization capabilities of microfluidics make it a good candidate for drug synthesis, drug delivery, formulation, and susceptibility testing [122,123]. Here, we discuss the integration of AI with microfluidics for drug design, synthesis, and formulation. The susceptibility testing application is discussed later in the following section.

Reinforcement learning provides an in silico means to generatively iterate drug design based on input physicochemical properties. Popova et al. utilized such an approach to create novel chemical libraries where the neural network can design chemicals by varying physical properties [124]. Recently, Grisoni et al. took a similar approach to generatively design drugs and further integrate with a microfluidic platform to synthesize the agonists against the liver X receptor [121] (Figure 3d). Such an automated platform combining AI and microfluidics introduces a closed-loop iterative optimization of drug design and synthesis that reduces human intervention. In the future, the improvement of algorithms with active learning could refine and optimize the design and synthesis process continuously. Similarly, machine learning used for iterating polymer synthesis and data-driven evaluation can accelerate drug formulation development [125].

## 4. Micro-Total Analysis System (μTAS) and Clinical Diagnostics

One major application of microfluidics is the automation of chemical analysis and clinical diagnostics, especially for point-of-care testing (POCT) [126,127]. In lab-on-chip technologies, various experimental processes are integrated and miniaturized onto a single microfluidic chip [128]. Further enhancement of data analysis and exclusion of potential human prejudice by integrating AI sets a feasible path toward realization of a μTAS [129].

### 4.1. Disease Diagnosis and Prognosis

To monitor human physiological status, detection and quantification of biomarkers using calibrated biosensors are required. Although conventional quantification by interpolation with a reference is standardized, utilization of AI to quantify and detect anomalies from pattern recognition in long-term clinical data presents unique opportunities. Combining paper microfluidics and ANN, the glucose concentration presented by the colorimetric method can be quantified and calibrated with high accuracy, as demonstrated in artificial urine and saliva [130,131].

Moreover, sensing applications based on image classification and recognition are particularly suitable for machine learning methods. Munoz et al. reported a machine learning method to detect label-free DNA based on fractal structures in subnanoliter droplets after loop-mediated isothermal amplification (LAMP) [132]. Similarly, optical detection of debris containing ambient RNA in droplets and excluded in data preprocessing can improve the data quality for single-nucleus RNA sequencing by using a semisupervised machine learning technique [133]. Alternatively, machine learning algorithms can also be applied to electrical inference of size and detection of ultrafine particles with low mean error [134].

Machine learning algorithms are sensitive to changes in cellular optical features due to biophysical properties or protein deformation. AI-enhanced microscopic image analysis of cells and tissues, therefore, is used for disease diagnosis in clinical settings. Deep-learning-assisted classification of the adhesion and deformation of RBCs in microchannels can be used to diagnose and monitor sickle cell anemia [135,136,137]. Rizzuto et al. combined microfluidics and AI-based video analysis to classify RBCs after passing microfluidic constrictions for diagnosis of hereditary hemolytic anemia [138]. The neural network can also predict the rigidity of RBCs directly from brightfield microscopy images. Ellett et al. used a microfluidic device to quantify the mobility of neutrophils in whole blood in combination with machine-learning-based scoring for diagnosis of sepsis with 97% and 98% sensitivity and specificity [139].

Similarly, rapid classification and quantification of cancer cells in various tissues and bodily fluids can be accelerated by AI in the fields of liquid biopsy or digital pathology, aiding clinicians and clinical laboratory scientists in disease diagnosis and prognosis, such as in leukemia, prostate, breast, brain, and lung cancers [140,141,142,143,144,145]. Many cells shed from primary tumors into patients’ circulatory systems in the form of circulating tumor cells (CTCs). The amount of CTCs in patients’ blood, even after treatments, correlates with the prognosis, also known as minimum residual disease (MRD). Therefore, concentration of CTCs is a novel biomarker for cancer management. The presence of CTCs can be analyzed and can predict prognosis with the aid of AI [146,147].

Mainstream immunochemical biosensors utilize the recognition ability of antibodies to detect and quantify biomarkers in various bodily fluids. Integrated with microfluidics, McRae et al. developed a compact programmable bionanochip platform (pBNC) on which multiplex immunoassays against prostate cancer, ovarian cancer, acute myocardial infarction, and drugs of abuse were performed on disposable cartridges for POCT applications. Afterward, the data were relayed to machine learning algorithms for further diagnosis based on previously trained data (Figure 4a) [148,149,150]. Furthermore, optical-based immunoassay readout can be resolved with the aid of machine learning. Song et al. reported a digital-enzyme-linked immunosorbent assay (ELISA) microarray for multiplex quantification of cytokines in patients who underwent cancer therapy [151] (Figure 4b). The turnaround time of the 12-plex assay was within 40 min by taking advantage of the microwell-based microarray and automated AI image analysis for biomarker quantification.

Immunochemical methods, however, suffer limitations such as interference to ligand–antibody interaction and the concentration-dependent hook effect, limiting the sensitivity, specificity, and linear range of detection [152]. Utilizing machine learning algorithms has also been shown to drastically improve the assay specificity and quantification of very high target concentration in surface-enhanced Raman spectrometric immunoassay and particle agglutination immunoassay [153,154,155].

In recent decades, the use of microfluidics for clinical testing has seen much growth due to increased adaption of POCT. At the same time, deep learning methods spur rapid development of AI across broad disciplines and applications. Humankind then met the challenge of a global pandemic at a scale not seen since the 1918 Spanish flu, the coronavirus disease 2019 (COVID-19), caused by severe acute respiratory syndrome coronavirus 2 (SARS-CoV-2) [156,157]. The rapid spread of the disease came with devastating consequences where healthcare systems were stretched to limits, state economies were stalled, and social norms were disrupted. Microfluidics was quickly adapted to develop quick turnaround of immunological as well as molecular antigen tests for rapid identification and diagnosis of exposed citizens and patients [158,159,160] for subsequent quarantine to curb the spread of the infectious disease. For example, Bhuiyan et al. developed an AI-controlled programmable microfluidic platform to perform ELISA immunoassays for cardiovascular disease diagnosis and SARS-CoV-2 diagnosis [161,162]. Many microfluidic POCT sensing platforms are also integrated with AI for analysis. Combining information and communication technologies (ICTs) with microfluidics, such as interfacing a smartphone with POCT devices, provide a solution for rapid testing in low-resource settings where gold-standard molecular diagnostics may not be easily available [163,164,165,166].

The combination of rapid testing offered by microfluidics and automated data analysis by AI has immense potential for proactive management of emerging infectious diseases. Not only can new diagnostics tests be rapidly developed after identification of antigen for future emerging infectious disease, but development of vaccines and drugs could be accelerated as well, through automation by microfluidics and AI [167,168,169,170].

**Figure 4 micromachines-14-00826-f004:**
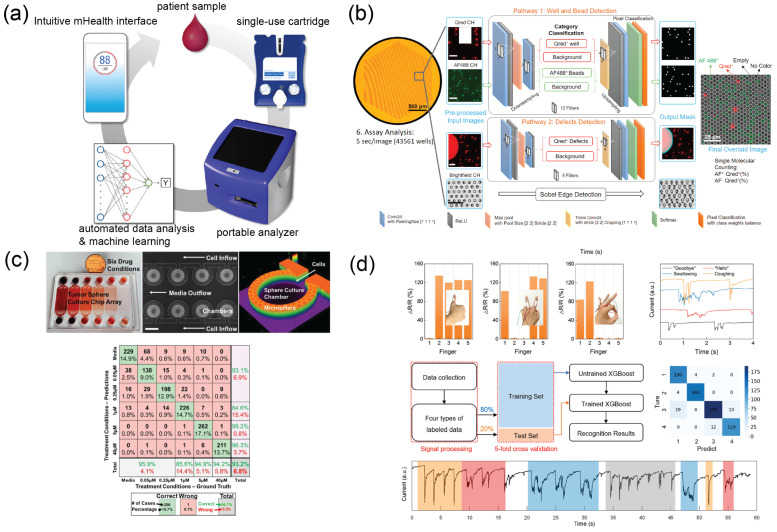
Micro- total analysis systems integrated with machine learning for clinical diagnostics. (**a**) A compact programmable microfluidic platform capable of performing multiplex immunoassay for POCT applications. Reprinted with permission from [150]. Copyright 2016 American Chemical Society. (**b**) A digital ELISA microarray for cytokine quantification with CNN-guided image processing. Reprinted with permission from [151] with permission from Elsevier. (**c**) A microfluidic platform with a convolutional neural network to predict drug efficacy based on the viability of cultured cancer spheres. Reprinted with permission from [171]. Copyright 2019 American Chemical Society. (**d**) A wearable sensor platform with AI data analysis for motion classification [172]. Reproduced under CC–BY license.

### 4.2. Drug Susceptibility Testing

Integrated microsystems can not only isolate target cells for analysis, but they also integrate with mixing functions to investigate the susceptibility of target cells to pharmaceutical agents, an important parameter for clinical decisions. One common application is the liquid biopsy. Kobayashi et al. showed that by combining microfluidic flow cytometry, time-stretch microscopy, and machine learning enables high-throughput, label-free analysis drug susceptibility of breast cancer, leukemia cells, and platelets in as little as a 500 μL sample [173,174,175].

Other major applications for drug susceptibility testing in clinical settings are therapeutic drug monitoring (TDM) and antimicrobial susceptibility testing (AST), where AI algorithms have begun to be adapted for monitoring and prediction [176,177]. Particularly for conventional AST, the time-to-result is of the essence for patient care. Therefore, automated microfluidic platforms in conjugation with optical detection and deep learning-powered analysis to rapidly detect bacteria, as well as screening and identifying appropriate antibiotics choices and therapeutic ranges, are driven by clinical needs [178,179,180,181].

Three-dimensional tissue models such as spheroids have the potential of mimicking tissue heterogeneity and drug transport in the microenvironment of an organism. However, 3D tissues have poor optical transmission in the visible light regions, making optical characterization and evaluation difficult, especially for conventional computer vision algorithms. Deep learning models learn the features of target structures through pattern recognition in images; therefore, they have great potential in image analysis for 3D tissue images. Zhang et al. developed an automatic microfluidic chip in combination with CNN-based image screening for evaluating drug susceptibility through viability prediction of tumor spheres (Figure 4c) [171]. The effective drug changes the morphology and optical quality of the spheroids, and AI models learn to associate the image with phenotypic results. Therefore, the AI models are capable of predicting phenotype based on label-free images. The trainability and predictive potential of deep learning in combination with microfluidic drug testing platform could be a promising alternative to animal testing [182].

### 4.3. Smart Wearables

Microsystems engineered through semiconductor processes are often rigid; therefore, they are limited from conformal sampling on organisms despite portability. When lab-on-chip systems are engineered on flexible substrates such as silicone or polyimides, such wearable microsystems enable long-term seamless biosampling and measurements [126] where conventional medical devices are often limited. Zhang and Tao showed that soft electrical sensors adhered on skin in combination with machine learning enabled long-term electrophysiological measurements [183]. In the management of cardiovascular diseases, early detection of abnormal electrical signals empowered by AI also has tremendous interests [184]. Similar needs in long-term monitoring of glucose in diabetes mellitus (DM) patients could also benefit from the combination of wearable devices and machine learning to predict the onset of hyperglycemia on the fly [185,186].

Soft sensors attached to the human body permit the measurement of pressure, force, and strain to infer biomechanics. Machine learning algorithms can be used to calibrate sensors [187,188] as well as tracking and predicting human gaits and motions (Figure 4d) [172,189]. Joining a multitude of sensors with Internet connectivity with AI at the core of data analysis offers a promising future in personalized healthcare [190].

## 5. AI Approach for Quantitative Biology

Machine learning algorithms can discern patterns in complex and noisy biological data; therefore, advances have been made recently in the field of cell analysis and personalized medicine by adopting AI for quantitative biology [191].

### 5.1. Cell Analysis

Counting, classifying, and sorting of cells are essential processes that have been long adopted in microfluidics and not only used in clinical diagnostics but also in basic research. Integration with AI further extends the data processing ability, such as prediction and pattern recognition for phenotype identification in cell data, which initiated a new era of intelligent microfluidics [192].

#### 5.1.1. Cell Counting and Classification

The number of suspended cells in a solution can be counted conventionally through optical detection, electrical resistance (Coulter method), or impedance [193]. Machine learning algorithms such as support vector machine (SVM), logistic regression, or ANN can extract cell features in optical images [194], lens-free holographic images [195], electrical data [196], or a combination of two [197] for counting and classification.

In combination with a brightfield or fluorescence microfluidic flow cytometer, cells can be counted, classified, and tracked at high speed using supervised learning as well as unsupervised learning models [198,199,200,201,202]. Heo et al. showed that a supervised learning pipeline combining a fully-connected regression network and CNN distinguished K562 leukemia cells and RBCs with mean average precision (mAP) as high as 93.3% at 500 FPS [198]. Deep learning models trained for virtually enhancing resolution can also be combined with low-resolution, high-speed microfluidic flow cytometry for counting and classifying algae, mammalian cells, and yeasts [201].

Minute optical property differences in cells which are difficult to discern by the naked eye can be classified by machine learning algorithms after proper training. Rossi et al. showed that CD4 and CD8 T lymphocytes can be differentiated by their scattered light due to biophysical property differences [203]. Polarization microscopy can also be conjugated with machine learning for the detection and classification of non-small-cell lung cancer cells without any staining [204].

In quantitative phase imaging (QPI) techniques, phase differences contributed by differences in refractive index of various cellular components can be quantified using digital holographic microscopy [205], lens-free holographic microscopy [206], interferometry [207,208], or time-stretched microscopy [209], both in 2D and 3D [210]. Conventional algorithms for phase retrieval are often time-consuming because image data can be noisy. Using machine learning, optical properties such as absorption, scattering, and refractive indexes can be used to extract biophysical information of cells such as dry mass. Without any labeling, only minute differences exist between nucleated white blood cells (WBCs) and cancer cells. Hirotsu et al. demonstrated that deep learning can assist label-free identification of cancer cells from WBCs, with the area under the curve of receiving operating characteristics (ROCs) reaching 95.7% [208]. Especially in time-stretched microscopy, broadband optical pulses image cells without any label at extremely high frame rates through photodiode readout and reconstruction [211,212,213,214,215]. Leukemia cells, RBCs, algae, and platelets can be classified by CNNs at extremely high flow speed (>10,000 cells per second) inside microfluidic flow cytometry with high accuracy (>96%) [216]. High-speed, time-stretched imaging can also couple with high-speed microfluidic cell classification and real-time label-free electrostatic cell sorting enabled by powerful GPU processing (Figure 5a) [213].

Electrodes can readily be fabricated with microfluidic channels and used for detection of cells by resistance or impedance detection. Machine learning algorithms can be further integrated into data analysis to classify objects based on their resistance or impedance characteristics [217,218,219,220,221]. Joshi et al. combined ink-jet printed microfluidics with machine learning algorithms to classify and differentiate three types of cancer cells based on their impedance profiles [217]. A deep learning algorithm with closed-loop feedback can regulate precise cell impedance measurement and classification [219]. Similarly, Feng et al. combined a microfluidic impedance flow cytometry with machine learning, showing capability of classifying five different cancer cells based on their membrane capacitance, cytoplasm, conductivity, and radius, with an accuracy of 91.5% [220].

#### 5.1.2. Cell Sorting

Flow cytometry is a cornerstone for high-throughput single cell classification with applications across clinical diagnostics and basic biomedical research. Further coupling the selection function for interested target populations provide many downstream applications, such as single cell molecular analysis, immunoassay, and selective cultivation [222].

Machine-learning-powered cell classification in brightfield and fluorescence flow cytometry can be integrated with a microfluidic flow cell with piezoelectric or pneumatic actuators to hydrodynamically collect cells or droplets of interest from classification in real time [97,223,224,225]. By adapting lightweight CNNs, high-speed electronics, and GPUs, real-time image classification and precise timing control for cell sorting have been realized. Aforementioned ultrahigh-speed QPI imaging systems integrated with sorting functions have been demonstrated, reaching accuracy above 94% at 100 events per second [226,227,228] (Figure 5b). By using machine learning to classify sperm and predict their DNA fragmentation index, a sperm sorter was developed that has broad applications in fertility medicine as well as in the veterinary industry [229].

Other cell sorting methods, such as magnetic sorting and dielectrophoretic cell sorting, have also been reported to integrate machine learning [230,231]. In particular, magnetic immunoseparation is a well-adapted concentration and purification method for clinical diagnostics as well as biotechnology. Uslu et al. showed that SVM algorithms can assist identification of leukemia cells captured with light-obstructing magnetic particles with a precision of 91.6% under 40× objective [230]. Moreover, magnetic particles are often hard to remove completely after the sorting is completed. By using machine learning for image analysis and classification, the residual amount of magnetic particles can also be quantified.

#### 5.1.3. Cell Phenotype Analysis

The possibility of deep learning is demonstrated by its ability to extract the features related to minute optical differences contributed by biological microstructures. Fluorescence images of subcellular structures can be predicted with CNNs from cell images taken under brightfield microscopy without any staining [232,233,234]. Style transfer from one staining to another staining is also proven [235]. Such in silico labeling can also be applied to high-speed QPI-based microfluidic flow cytometry [236]. Moreover, deep learning methods have demonstrated high-quality segmentation, and they can further extract features regarding phenotypes.

The mechanical deformability and relaxation of cells can be probed by microfluidic shear flow or microstructure constriction [237,238,239]. By further applying analysis empowered with machine learning, the mechanical properties of cells can be predicted from optical images with accuracy reaching over 90% [137,240].

The health state of cells in microfluidic platforms for drug screening can also be predicted and have vast applications in drug susceptibility testing, immunotherapy efficacy testing, and drug resistance monitoring [241,242,243]. For example, to maintain cellular homeostasis, the timing and frequency of cellular metabolism, DNA duplication, and division are tightly regulated in the cell cycle in four phases: gap 1 (G1), synthesis (S), gap 2 (G2), and mitosis (M). Conventionally, identification of the cell cycle requires staining of cells against molecular markers for nucleic acid and various essential proteins for DNA replication [244]. By training CNNs on training data annotated from fluorescence information, the CNNs can be used to classify cell cycles and even predict them in brightfield images [245,246,247,248]. With the combination of microfluidic trapping, such a platform can have broad applications for predicting cell cycle phenotypes in drug screening and basic research [249,250]. Aspert et al. used microfluidic and deep learning for high-throughput tracking of single cell division and emergence of mutants [250] (Figure 5c). By adapting deep learning, analysis of the survival and phenotype changes of large amounts of single cell data could be automated, greatly saving time, minimizing human bias, and increasing statistical power.

Cellular senescence upon exposure to reactive oxygen species, a key contributor to aging, is also a phenotype that requires quantification by staining, typically against senescence-associated beta-galactosidase activity. CNNs have also been demonstrated to predict cellular senescence based on the morphology or optical quality of the cell in brightfield [251,252]. By expanding the AI for phenotype prediction, cellular phenotypes in biomimetic microenvironments established by microfluidics have broad potential applications.

#### 5.1.4. Spatiotemporal Cellular Dynamics Analysis

Deep learning methods, especially CNNs, are particularly effective in image segmentation across brightfield and fluorescence microscopy, outperforming conventional computer vision methods [253,254,255]. Accurate segmentation accompanied by precise tracking enables the accurate analysis of complex cellular dynamics in a stack of timelapse images and contributes to quantitative understanding of cellular processes. For example, Bove et al. used CNN classification and Bayesian inference tracking for understanding the role of cell competition in epithelial tissue maintenance [256]. Accurate segmentation by deep learning also benefits tracking cell migration in both 2D and 3D [257,258,259,260].

Through coupling with microfluidics, cellular dynamics can be studied in detail in dedicated microenvironments compatible with high-resolution imaging. Single bacterium evolution and bacteria population dynamics can be experimentally modeled on microfluidic devices and analyzed using machine learning models [261,262,263,264].

**Figure 5 micromachines-14-00826-f005:**
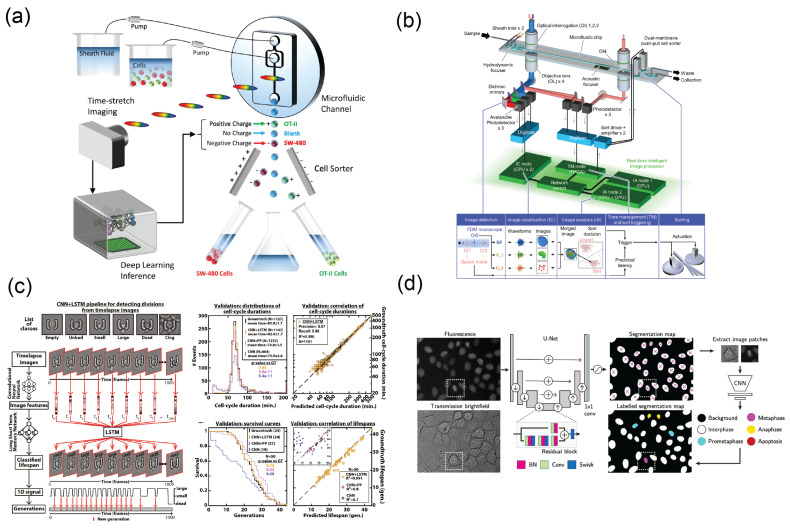
Microsystems integrated with AI for cell analysis. (**a**) An optofluidic time-stretch QPI cell sorter based on high-speed, label-free cell classification and sorting empowered by deep learning [213]. Reproduced under CC–BY license. (**b**) Frequency-multiplexed microscopy coupled with real-time machine learning processing and piezoelectric actuator for high-speed cell sorting. Reprinted from [226] with permission from Elsevier. (**c**) A microfluidic cell trap with AI phenotype prediction for yeast division tracking and survival analysis [250]. Reproduced under CC–BY license. (**d**) A deep learning framework to automatically classify cell cycles and track lineage trees [265]. Reproduced under CC–BY license.

Wang et al. used a deep learning-based segmentation with Bayesian filtering to track migration of cells of a 3D angiogenic vessel with an accuracy of 86.4% [266]. Cell migration guided by physical or chemical gradients can be studied with high-throughput microfluidic platforms, while AI-empowered tracking relieves humans from the tedious task. Quantification of cell migration and growth kinetics can be analyzed with reduced risk of human bias [267,268]. Furthermore, accurate cell tracking across multiple cell divisions enables building of lineage trees, and is capable of revealing the population dynamic information of cells throughout the experimental period. Ulicna et al. combined residual U-net, CNN classification, and Bayesian tracking to reconstruct the cellular lineages in thousands of hours of timelapse data and identified cell cycling heterogeneity and correlated cyclings between cells of similar generation [265] (Figure 5d). Similar machine learning inference can also be applied to resolving temporal dynamics of multicellular organoids at single-cell level in 3D [269,270].

### 5.2. Personalized Medicine

Advances in quantitative biology in modern biomedical research spur the development of personalized medicine, a discipline where biological variation is recognized and tailored care is aimed towards every individual [271]. Microfluidics and AI as an integrated solution has great potential and opportunity for personalized medicine.

Automated microfluidics and POCT devices can be deployed for dedicated patients for rapid turnaround of personalized molecular diagnostics results. Moreover, patient-derived cells, tissues, and induced pluripotent stem cells (iPSCs) can be incorporated on a microfluidic platform, also known as an organ-on-chip (OoC), recreating patient-specific multicellular microenvironments [272]. The OoC has broad potential, from being a clinical diagnostics vehicle to a personalized drug development tool as well as a fundamental biomedical research platform. By using human- or patient-derived cells and tissues, the results are more clinically translatable, and AI-empowered analysis can deliver prediction of prognosis and therapy outcome [273]. Furthermore, the OoC could potentially replace animal studies, resolving decades-long dilemmas in research ethics [274,275].

#### 5.2.1. Integration with Molecular Bioinformatics

Microfluidic technologies have been adapted for genomics and transcriptomics studies, both in the core of the sequencing platform, such as sequencing flow cells of next-generation sequencing (NGS), and sample and library preparation, such as single cell isolation using droplet microfluidics [276,277]. Because many of the techniques use Poisson statistics, by adopting machine learning algorithms in bioinformatic pipelines, the quality of library preparation and data quality can be improved and helps advance multiomics studies such as spatial transcriptomics and metabolomics [191,278,279]. For example, Ko et al. applied nanofluidics and machine learning to predict cancer in mice and patients using exosomal mRNA profiles learned through linear discriminant analysis [280].

#### 5.2.2. Organ-on-Chips as Human Mimetic Models

Organ-on-chips (OoCs) aim to recreate multicellular complex environments in vitro on microfluidic platforms to recapitulate the complex tissue interaction with physical and chemical gradients existing in the biological microenvironment [281]. Many OoCs adapting human-derived primary cells, stem cells, or cell lines have been developed for different organs models, such as brain [282,283], thyroid [284], heart [285], stomach [286], intestine [287], lungs [288,289], liver [290], kidneys [291], pancreas [292], and bladder [293].

The complexity of OoC platforms offers many opportunities for automation and data analysis, both of which can benefit a lot from AI [294]. In particular, when patient-specific tissues or iPSC-differentiated cells are used, OoCs have the potential to recapitulate the human physiology and deliver personalized medicine without the need for animal testing [295,296,297].

Due to the excellence in image segmentation and predictive potential, AI has started to be integrated with OoC platforms for image analysis, correlation, and prediction of clinical results. Paek et al. created a bone OoC as a drug-testing platform for osteoporosis. CNN-based image segmentation provided quantitative measurements of beta-catenin translocation to evaluate the efficacy of the osteoporosis suppressive antibody [298] (Figure 6a). Likewise, a blood–brain niche (BBN) chip was developed to model the extravasation of breast cancer cells into a brain-mimetic niche [299,300]. The invasion of cancer cells through an endothelial cell sheet was evaluated by confocal 3D tomography, and several machine learning algorithms were benchmarked to classify the metastatic potential of the cancer cells. The ANN was found to outperform other machine learning algorithms (Figure 6b). Chong et al. developed a microfluidic multicellular coculture array for testing skin sensitivity to drugs [301]. By combining SVM and principle component analysis on image results for several apoptosis markers, adverse cutaneous drug reactions can be predicted with the accuracy of 87.5% and the specificity of 75%. The predictive evaluation could have huge potential in personalized medicine to aid in preventing severe adverse reactions, such as Steven Johnson syndrome and toxic epidermal necrolysis syndrome.

Alternatively, a sheet of muscle cells placedon a microfluidic stretching platform can be quantitatively studied with the assistance of AI. Recurrent neural networks (RNNs) with long short-term memory (LTSM) blocks were used to predict the subsequent dynamics of muscle cells from one time point in conjunction with a CNN to predict the phenotype of them [302] (Figure 6c).

The complexity of the immune-tumor niche also makes it a key topic suitable for combined approach by OoC and AI analysis [303]. The interaction between dendritic cells and leukocytes with cancer cells in a 3D microfluidic coculture platform can be accurately segmented and tracked with the aid of deep learning [304,305,306,307]. Nguyen et al. constructed an immune-competent OoC that reconstitutes cancer, immune, endothelial cells, and fibroblasts altogether for studying immune–tumor interactions and immunomodulation of cancer-associated fibroblasts [306] (Figure 6d). The synergy between OoC and AI shows a promising future of personalized medicine.

#### 5.2.3. Personalized Drug Development

One key concept of personalized medicine is recognizing that individuals have biological variations both in biochemistry and drug responses [308]. Utilization of AI for personalized drug optimization by both drug selection and dosage has been promising [309,310]. A microfluidic POCT device for MRD and an OoC for tracking cancer migratory abilities were demonstrated for evaluation of patient-specific cancer drug efficacy by combining AI analysis [311,312]. It is expected that we will see more examples and applications combining microfluidic biomimetic models integrated with AI analysis and prediction in the future, such as integrated lab-on-chip devices for personalized diagnosis, medicine formation, and even drug delivery [313].

**Figure 6 micromachines-14-00826-f006:**
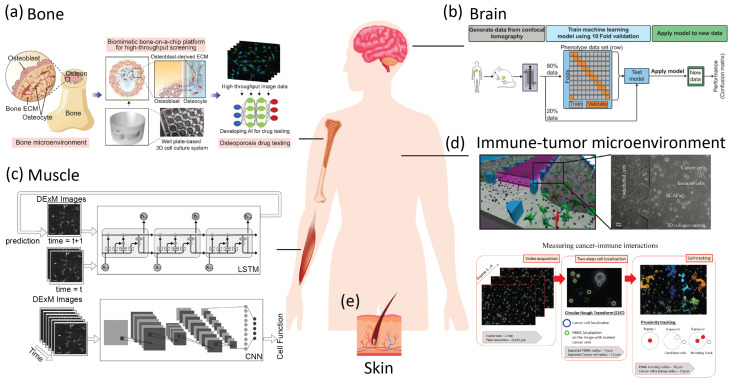
Organ-on-chip platforms integrated with AI analysis. (**a**) A bone-on-chip for osteoporosis drug testing and development [298]. Reproduced under CC–BY license. (**b**) A blood–brain niche OoC for predicting the metastatic probability of brain cancer [299,300]. Adapted with permission from [300]. (**c**) A muscle-on-chip with AI image analysis for quantitative understanding of muscular cell phenotypes. Reprinted with permission from [302]. (**d**) An OoC to reconstitute immunocompetent tumor microenvironment in breast cancer [305,306]. Reproduced under CC–BY license. (**e**) A skin-on-chip for predicting adverse cutaneous drug reactions [301]. Components in the figure were designed by macrovector; brgfx/Freepik.

## 6. Discussion and Outlook

Microfluidics is a rapidly advancing field that involves manipulating small fluid volumes at the microscale. One of the key challenges in this field is the ability to control and analyze the complex fluid dynamics that occur on such a small scale. In recent years, there has been growing interest in integrating AI with microfluidics to address this challenge. AI has the potential to revolutionize microfluidics by enabling the development of intelligent control systems that can adapt to changing conditions in real-time. Additionally, AI-based image analysis techniques can automatically detect, classify, and track objects of interests in microfluidic systems, providing valuable insights as well as prediction of their dynamics. The integration of AI with microfluidics could lead to the development of more sophisticated lab-on-a-chip diagnostic devices, more efficient and effective drug delivery systems, and more versatile monitoring platforms. AI-empowered image analysis delivers quantitative analysis of cells and tissues in complex OoCs and opens a new era in quantitative biology and personalized medicine.

Overall, the advantages of integrating AI with microfluidics include automation, efficiency, and quantitative analysis in data processing, as well as cost effectiveness due to the low reagent consumption of microfluidics. AI can perform large-scale iterations and automation of planning and executing experimental parameters. The trainability of AI empowered it to flexibly learn from data and predict even when a priori annotation is not always available. The neural networks especially outperform conventional algorithms in image and language recognition, which are often abundant in biomedical and clinical data, opening possibilities to automate quantitative analysis of data that was conventionally difficult to analyze.

However, some challenges must be addressed. Firstly, robust AI algorithms and reliable data inputs are necessary for successful integration. While recent advancements have demonstrated the power of AI algorithms, they are often considered a “black-box” due to their lack of explainability. To build trustworthy AI-integrated microfluidic systems, it is crucial to prioritize explainability and validation because quality assurance and risk management are of the utmost importance in medical instrumentation. Secondly, AI operations can be computationally expensive; both training and interference demand immense parallel processing power. Therefore, energy efficiency and deployability for particular applications and their environments are essential. For example, when integrating with a microfluidic POCT system in a low-resource setting, such as a rural area with intermittent power supply, a lightweight AI on an edge computing platform may be crucial for successful integration. Additionally, data size is a critical factor in AI performance, which scales proportionally with the amount of the data. While combining the automation capability of microfluidics and MEMS with data from existing methods for training AI systems can be challenging, data fusion could provide a unified and comprehensive dataset that enables AI systems for more reliable predictive analysis. Cooperation within the scientific community for data sharing and training AI systems with federated learning can also address the challenges of both explainability and deployability.

In summary, by integrating the compact and versatile microfluidic platforms with the intelligent control and analysis with AI, we expect new breakthroughs in healthcare in the near future, but caveats must be taken to validate the AI system for any applications and evaluate potential risks.

## Figures and Tables

**Figure 1 micromachines-14-00826-f001:**
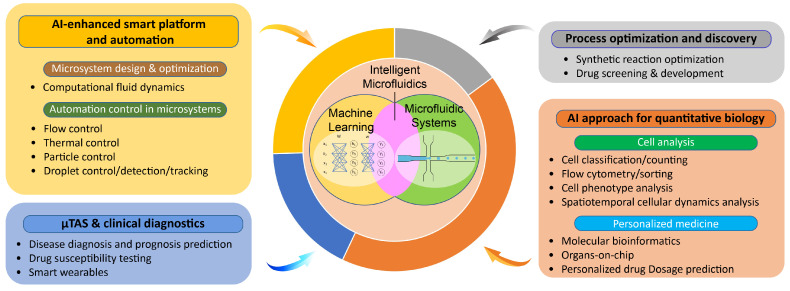
Diagram of microfluidic applications integrated with artificial intelligence (AI). μTAS stands for micro-total analysis systems.

## Data Availability

No new data were created or analyzed in this study.

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
