# Peer review of "Microsystem Advances through Integration with Artificial Intelligence"

_micromachines, 2023, doi:10.3390/mi14040826_

Round 1

Reviewer 1 Report

In this article, Tsai and colleagues review the advances enabled via artificial intelligence for the microsystems. This review discusses intelligence microfluidics in four sections, AI-enhanced smart platform and automation, microTAS & clinical diagnostics, process optimization and discovery, and AI approach for quantitative biology. The review article looks fine and is well organized. The references are sufficiently described and are relatively up-to-date. However, when reading through the four sections of the article and after reviewing around 300 references, the reviewer expects to see a much more enriched discussion and future outlook for the current article. So based on the proposed structure of the review and the detailed explanations, what should the reader expect as the future perspective of the field? What are the challenges, and what is the authors' opinion on addressing those issues? This requires proper attention and need more enrichment. Also, the next point being mentioned is that after the recent pandemic, there has been a surge of interest in the use of microfluidics and artificial intelligence for COVID-19. The reviewer highly encourages the authors to add these articles and pay special attention to this topic in the case of the recurrence of the pandemic. Finally, there has been some report on the use of artificial intelligence for designing microfluidic mixers, which is not well covered in the article.

Addressing these comments will enrich the article and make it appropriate for publication in the journal of Micromachines. 

Reviewer 2 Report

This review summarized key microfluidic advances integrated with AI and discuss the outlook and possibilities of combining AI and microfluidics. Deployment of smart microfluidics may be tremendously beneficial in various applications in future including high-throughput drug discovery, rapid point-of-care-testing (POCT), and personalized medicine. The authors have done very thorough research, classified clearly, and the summary made is meaningful and informative. I would accept the article after the following comments are address.

1. Is there anything missing after "and" in the last sentence of part 2.2.1, line 108?

2. There are too many small paragraphs in the text, especially it often appears that one sentence is used as a paragraph. For example, at line 221, could the first paragraph of part 3.2 be considered to be combined with the next paragraph?

3. The selected images should be typical cases of the presented content. However, some pictures are introduced too briefly, and it is recommended to supplement the introduction of certain pictures, such as Figure 2a. Please highlighting the features and advantages of the main cases.

4. The final discussion section is too small and not specific enough to summarize. And it should list the current shortcomings and specific outlook.

Reviewer 3 Report

In this review, the key microfluidic advances integrated with artificial intelligence (AI) have been summarized by the authors. In addition, the outlook and possibilities of combining AI and microfluidics are discussed. However, there are some problems in the review. 1. The Reference is missing in the sentence “For example, Wang at al. adapted a fully automated convolutional neural network (CNN) computer vision to aid in calibration during 3D printing of microstructure dimensions.” 2. In Figures 2, 3, 4, 5 and 6, the text in these figures is not clear. 3. I recommend to present a summary table that explains advantages or limitations of specific models or approaches.

Round 2

Reviewer 1 Report

The authors successfully addressed all my comments.